# The Impact of Selected Bacterial Sexually Transmitted Diseases on Pregnancy and Female Fertility

**DOI:** 10.3390/ijms22042170

**Published:** 2021-02-22

**Authors:** Katarzyna Smolarczyk, Beata Mlynarczyk-Bonikowska, Ewa Rudnicka, Dariusz Szukiewicz, Blazej Meczekalski, Roman Smolarczyk, Wojciech Pieta

**Affiliations:** 1Department of Dermatology and Venereology, Medical University of Warsaw, 02-008 Warsaw, Poland; ksmolarczyk@gmai.com (K.S.); beata.mlynarczyk@wum.edu.pl (B.M.-B.); 2Department of Gynaecological Endocrinology, Medical University of Warsaw, 02-091 Warsaw, Poland; ewa.rudnicka@poczta.onet.pl; 3Department of General and Experimental Pathology, Medical University of Warsaw, 02-091 Warsaw, Poland; dariusz.szukiewicz@wum.edu.pl; 4Department of Gynaecological Endocrinology, Poznan University of Medical Sciences, 60-535 Poznan, Poland; blazejmeczekalski@yahoo.com

**Keywords:** STI, pregnancy, fertility, resistance mechanisms, genotyping

## Abstract

Sexually transmitted infections (STIs) caused by *Neisseria gonorrhoeae*, *Chlamydia trachomatis* and *Mycoplasma genitalium* are a common cause of pelvic inflammatory disease (PID) which can lead to tubal factor infertility (TFI). TFI is one of the most common causes of infertility, accounting for 30% of female fertility problems. STIs can also have an impact on pregnancy, leading to adverse pregnancy outcomes. Escalating antibiotic resistance in *Neisseria gonorrhoeae* and *Mycoplasma genitalium* represents a significant problem and can be therapeutically challenging. We present a comprehensive review of the current treatment options, as well as the molecular approach to this subject. We have given special attention to molecular epidemiology, molecular diagnostics, current and new treatments, and drug resistance.

## 1. Introduction

Infection agents such as bacteria, viruses, and fungi can impair various human functions, including reproduction and pregnancy. Among the most common microorganisms which cause sexually transmitted diseases are *Chlamydia trachomatis*, *Neisseria gonorrhoeae*, and, to a lesser extent, *Mycoplasma genitalium* [1,2,3]. Infection can ascend from the vagina, through the cervix, to the upper genital tract, the endometrium, and ultimately to the fallopian tubes and clinically presents as acute pelvic inflammatory disease (PID) [3]. Approximately 15% of women with PID then develop tubal factor infertility (TFI) and the number of episodes of PID is directly proportional to the risk of infertility [4,5]. TFI ranks among the most common causes of infertility, accounting for 30% of female fertility problems [6].

Microorganisms such as *C. trachomatis* and *N. gonorrhoeae* can not only affect tubal patency but are also associated with pregnancy complications, including ectopic pregnancy, recurrent pregnancy loss, and preterm birth [7]. In addition, other pathogens such as *M. genitalium* can also play a role in pregnancy complications [7,8].

Infections caused by these pathogens are common around the world and have a great impact, as mentioned, on reproductive health. Recovering from the disease does not give immunity. Therefore, timely diagnosis and timely treatment are very important. The spreading of the drug resistance of *N. gonorrhoeae* and *C. trachomatis* necessitates the development of new drugs and treatment regimens.

This paper presents the latest data on molecular diagnostics, current and new treatments, and drug resistance.

## 2. Epidemiology

### 2.1. General Epidemiology (Including Epidemiology in Women and/or Pregnant Women)

Each year, there are an estimated 357 million new cases worldwide of four curable sexually transmitted infections (STIs) among people 15–49 years of age: *C. trachomatis* is acquired by 131 million people, *N. gonorrhoeae* is acquired by 78 million people, syphilis by 5.6 million people, and trichomoniasis by 143 million people [9]. Gonorrhea, one of the most common sexually transmitted infections (STIs), continues to be a major public health concern worldwide. In 2016, there were about 87 million new cases of gonorrhea; 20 cases per 1000 population were in women, and 26 cases per 1000 population were in men (15–49 years of age) worldwide [10]. The rates of cases of gonorrhea reported in the USA increased by 75.2% between 2009 and 2017, from a historic low of 98.1 cases per 100,000 population in 2009 to 170.6 cases per 100,000 population in 2017 [11]. China suffered a dramatic increase of 38.5% in *N. gonorrhoeae* infections, from 100,245 cases reported in 2015 to 138,855 cases in 2017, following over 10 years of sustaining a stable, or even slightly decreased, incidence with 109,525 cases each year [12]. With the exception of lymphogranuloma venereum (LGV), chlamydial infections are widely diffused among the general population, mainly affecting young people 16–24 years of age. Risk factors include high frequency of changing partners, multiple partners, unprotected sex, and being unmarried [13]. In the USA in 2018, 1,758,668 cases of chlamydia were reported to the CDC, which represents a 19% increase since 2014 [14]. For 2018, 26 EU/EEA Member States reported 406,406 confirmed cases of chlamydia infection (https://www.ecdc.europa.eu/sites/default/files/documents/AER-for-2018-STI-chlamydia.pdf (accessed on 2 November 2020)) To our knowledge, the population prevalence of *M. genitalium* has not been ascertained systematically. Nonsystematic reviews have reported prevalence estimates ranging from 0.7% to 3.3% in the general population and from zero to 20% in a range of female study populations described as “low risk” [15]. There is a limited amount of data regarding the rates of STIs during pregnancy. Analysis shows the incidence rate of chlamydia was 6.7 per 100 person years (py) and 9.9/100 py during pregnancy; the incidence rate of gonorrhea was 2.7/100 py and 4.9/100 py during pregnancy. Prevalence among pregnant women differs depending on the regions. In a systematic review, *N. gonorrhoeae* ranged from 1.2% in Latin America to 4.6% in Southern Africa, and *C. trachomatis* ranged from 0.8% in Asia to 11.2% in Latin America.

### 2.2. Genotyping and Molecular Epidemiology

*N. gonorrhoeae* genotyping is usually based on one or more of four methods: WGS (Whole Genome Sequencing), NG-MAST (*Neisseria gonorrhoeae* multiantigen sequence typing), MLST (Multi-locus Sequence Typing), and NG-STAR *(Neisseria gonorrhoeae* Sequence Typing for Antimicrobial Resistance). WGS enables the classification of investigated strains into related groups of so-called clades. As there is currently no worldwide database of *N. gonorrhoeae* clades (at the end of 2020) it is almost impossible to compare the strains in a wider perspective. However, WGS can help to simultaneously identify the NG-MAST, MLST, and NG-STAR sequence types of the bacteria.

NG-MAST is the most utilized method of *N. gonorrhoeae* genotyping. The method involves sequencing the 490 bp fragment of the *porB* gene encoding porin B and the 390 bp fragment of the *tbpB* gene that encodes the B unit of the transferrin-binding protein. The NG-MAST worldwide database makes it possible to determine the allele number of each investigated gene. About 12,800 *porB* alleles and about 3200 *tbpB* alleles were described by the end of 2020. NG-MAST types are established in the database of the *porB* and *tbpB* allele combination. About 22,000 NG-MAST sequence types were described by the end of 2020. About 30% of identified NG-MAST types are classified in a few hundred genogroups. In most European countries, the predominant genogroups are: G1407 (often associated with antibiotic resistance), G2992, G225, G51 G21, G387, G4995, G7445, G292, and G2400. Affiliation to some genogroups is often correlated with spread of the antimicrobial-resistant variants. High-level ceftriaxone-resistant *N. gonorrhoeae* strains belong to G1407, G564, G1791, G1866, G4019, G5267, G11018, and G11110. The high-level resistance to ceftriaxone and azithromycin together occurred in an *N. gonorrhoeae* strain classified as G1866. However, the prevalence of the G1407 genogroup has decreased recently in Europe and in some countries, such as Russia, the genogroup practically does not occur [16,17,18,19]

MLST has been used for a long time for the genotyping of many species of bacteria and some fungi too. MLST typing of *N. gonorrhoeae* is based on the sequencing of 450–500 bp fragments of seven genes involved in basic metabolism: *abcZ, adk, fumC, gdh, glnA*, *gnd*, and *pyrD.* The allele number of each gene can be determined in a public worldwide database and the sequence type is defined in the database of the seven-allele combination. The most prevalent *N. gonorrhoeae* MLST types in the European Union are: ST1901, ST9363, and ST7363 [18]. High-level ceftriaxone-resistant *N. gonorrhoeae* strains belong to ST1901, ST1903, ST7363, ST12039, and ST13637. *N. gonorrhoeae* simultaneously resistant to ceftriaxone and azithromycin was classified as ST12039 [16,17,18,19].

NG-STAR is a method that has been used since 2017 and is based on sequencing fragments of seven genes associated with beta-lactam, fluoroquinolone, and macrolide resistance: *penA* (which encodes PBP2)*, mtrR* (which encodes the repressor protein of the membrane pump *mtrCDE), porB* (which encodes porin B)*, ponA* (which encodes PBP1)*, gyrA* (which encodes a subunit of gyrase)*, parC* (which encodes a subunit of topoisomerase IV)*,* and the 23S rRNA gene (which contains a target site for azithromycin). The numbers of the alleles of the genes can be checked in public world databases and the NG-STAR type is determined on the basis of the seven-allele combination. By the end of 2020, about 3200 NG-STAR types as well as several hundred clonal complexes (CCs) were described [20]. NG-STAR seems to be the method best correlated with antimicrobial resistance. The high-level ceftriaxone-resistant *N. gonorrhoeae* strains belong to ST16 (CC90), ST133 (CC38), ST139 (CC139), ST226 (CC348), ST227 (CC348), ST233 (CC199), ST996 (CC73), and ST1133 (CC199). *N. gonorrhoeae* simultaneously resistant to ceftriaxone and azithromycin was classified as ST996 [16,17,18,19,20].

Types among *C. trachomatis* isolates are determined on the basis of a single-nucleotide polymorphism in the gene encoding the bacterial major outer membrane protein (MOMP) *ompA.* The investigation of *C. trachomatis* serovars/genovars is important not only from the epidemiological point of view but also because they have different pathogenic potentials. A, B, Ba, and C types cause trachoma, D-K, Da, Ia, and Ja cause urethritis and cervicitis, and L1–L3 and their subvariants, such as L2a and L2b, cause LGV. Most of the urogenital strains in Europe and worldwide belong to the E and F genotypes [21,22]. The most common LGV variant in Europe is L2b, but LGV is extremely rare in European women and mainly occurs in men who have sex with men (MSM) [23].

Four different schemes of the MLST method based on the analysis of fragments of seven genes have been used to genotype *C. trachomatis*. The data making it possible to determine genotypes are accessible in public databases for molecular typing and microbial genome diversity, *Chlamydiales*. The methods have a high discriminatory power; however, various schemes can create some problems when comparing the results. One of the most used schemes is based on the *glyA, mdhC, pdhA, yhbG, pykF, leuS*, and *lysS* genes. The most common sequence types are ST39 and ST34, which are associated with genotypes E and F [21,24].

The WGS of *C. trachomatis* strains makes it possible to determine four main clades: ocular, urogenital T1 (which involves the most common genovars, E and F), urogenital T2 (a less prevalent genovar), and LGV [25].

*M. genitalium* genotypes are determined on the basis of the *mgpB* gene. In the genotyping of strains from MSM in Germany, the most common types were 4, 6, 113, and 108 [26,27].

The data concerning WGS of *M. genitalium* genotypes are still incomplete. The WGS of 28 *M. genitalium* strains from different geographic regions makes it possible to determine that there are two main clades. Single-nucleotide polymorphisms in the *parC* and *23SrRNA* genes were associated with fluoroquinolone and macrolide resistance, respectively [28].

## 3. Pathogenesis (in Women)

### 3.1. The Pathogenesis of Fallopian Tube Damage

Among the most common microorganisms involved in the pathogenesis of tubal damage are *C. trachomatis*, *N. gonorrhoeae*, and *M. genitalium* [1]. *C. trachomatis* remains the most common bacterial cause of tubal obstructions, lacerations, and formations of adhesions which may disrupt the passing of oocytes through the tubes [4,6,10]. The serovars D-K of *C. trachomatis* have an affinity to the epithelial cells of the urogenital tract, migrating from the cervix to the uterus and the fallopian tubes, causing chronic inflammation [29]. The increased amount of heat shock protein (HSP; cHSP10 and cHSP57/60) synthesized by *C. trachomatis* also induces a proinflammatory immune response in the human fallopian tube epithelia, resulting in scarring and tubal occlusion [30,31]. The link between tubal factor infertility and previous *C. trachomatis* infection is so strong that the chlamydial antibody titer is commonly used in clinical practice as a surrogate marker for tubal factor infertility [30]. It should also be mentioned that *C. trachomatis* infections are commonly asymptomatic, implying a pathogenic strategy for the evasion of innate inflammatory immune responses. The outer membrane of *C. trachomatis* contains lipopolysaccharide (LPS), a known potent agonist of inflammatory innate immunity. Yanget al. found that *C. trachomatis* LPS is not capable of engaging the canonical TLR4/MD-2 or noncanonical caspase-11 inflammatory pathways [32]. The inability of *C. trachomatis* to trigger an innate immunity inflammatory response may explain the high incidence of asymptomatic chlamydial genital infection.

Primary *N. gonorrhoeae* infection is present in the endocervix often with concomitant urethral infection. Ascending infection may occur in 10–20% of infected women and can result in acute PID that can manifest as salpingitis, endometritis, and tubo-ovarian abscess, leading to sterility, ectopic pregnancy, and chronic pelvic pain [33]. Cytokines IL-1α, IL-1β, and TNF-α, produced by tubal epithelial cells after being challenged by *N. gonorrhoeae*, may contribute to the development of gonococcus-induced infertility [34]. During initial infection, following initial host cell interaction, *N. gonorrhoeae* attachment and subsequent colonization depends on type IV pili forming microcolonies on the epithelial cell surface. Adherence to the epithelial surface and subsequent pilus retraction bring the gonococci close to the cell surface [35]. In particular, *N. gonorrhoeae* attacks the epithelial cells of the fallopian tube, both initially by attaching to the nonciliated mucosal cells and by sloughing off ciliated mucosal cells. The resulting damage to the fallopian tubes hinders the transportation of the ovum along the tubes, elevating the risk of infertility and ectopic pregnancy [36].

While *N. gonorrhoeae* and *C. trachomatis* are known to be pathogens in salpingitis and tubal infertility, in many cases, other microorganisms are identified, i.e., *M. genitalium*. *M. genitalium* is strongly associated with cervicitis, endometritis, salpingitis, and PID, and in many cases can be the cause of infertility [37]. Antibody tests were positive in patients with tubal factor infertility, even in cases where chlamydiosis was excluded [37]. McGowin et al. found that the organism can attach to the epithelial cells of the reproductive tract and can induce cellular immune responses that result in inflammation [38,39]. In short, adhesion of *M. genitalium* to the host epithelial cells induces an acute inflammatory response via highly expressed immune sensors including Toll-like receptors 2 and 6. The binding of these receptors to *M. genitalium* and its lipoproteins results in NF-ĸB activation and acute induction of the genes involved in host defense. Those proinflammatory signals include potent chemokines that ultimately result in leukocyte recruitment to the site of infection. In addition to leukocytes, *M. genitalium* induces very potent proinflammatory responses from monocytes/macrophages, that are common to female reproductive tract tissues [40]. In another in vitro study by Baczynska et al., it was demonstrated that *M. genitalium* adhered to the epithelium of the human fallopian tube, causing the swelling of the cilia and the separation of the cilia from the epithelium [41]. In comparison with infections caused by *C. trachomatis* and *N. gonorrhoeae*, the damage to the fallopian tubes caused by *M. genitalium* tends to be moderate, but when left untreated, it may accumulate and yield serious long-term consequences for fertility [38].

### 3.2. The Pathogenesis of Infections during Pregnancy

Microorganisms like *C. trachomatis* and *N. gonorrhoeae* can not only affect tubal patency, but they are also associated with pregnancy complications, including ectopic pregnancy, recurrent pregnancy loss, and preterm birth [7]. The chlamydial 10 kDa and 57 kDa HSPs (cHSP10 and cHSP57/60) have a similar structure to human proteins and there is cross-reactivity between the human HSP60 and the bacterial cHSP60, which leads to the formation of antibodies against the HSP60. It has been found that these antibodies have a negative impact on embryonal growth and increase the probability of adverse pregnancy outcomes such as repeated implantation failure and recurrent miscarriage [42]. Equils et al. established that cHSP57/60 induces trophoblast apoptosis by stimulating Toll-like receptor 4 (TLR-4), which is responsible for the immune response in the placenta [43].

*C. trachomatis* and *N. gonorrhoeae* infections have also been linked to other adverse pregnancy outcomes, including chorioamnionitis, placentitis, premature rupture of the membranes, and preterm birth. Vertical transmission from the genital tract can also cause conjunctivitis and pneumonitis in newborns [7].

The role of *M. genitalium* in maternal infections and its impact on the outcome of pregnancy has been studied less and the evidence that the pathogen is associated with pregnancy complications is limited [7]. According to the last meta-analysis by Lis et al., *M. genitalium* infection has been shown to be significantly associated with an increased risk of spontaneous abortion and preterm birth in some studies, but evidence is inconsistent and further research is needed to consolidate the conclusions [44].

## 4. Symptoms in Women with Special Attention to the Impact on Pregnancy and Infertility

### 4.1. Neisseria gonorrhoeae

*N. gonorrhoeae* infection can be either symptomatic or asymptomatic in women (up to 50% infections can be asymptomatic). Symptoms are also less prominent if the infection affects the throat or the anal area. The most common manifestations of gonococcal infection include altered vaginal discharge, pain of the lower abdomen, or pain during sexual intercourse (which is associated with cervicitis), as well as dysuric symptoms (pain, burning sensation, itching, stinging, tingling when urinating or soon after), and occasional intermenstrual bleeding or menorrhagia [45].

The first symptoms usually occur 3–7 days after the intercourse (usually within the first week). Possible complications due to infection in women include disseminated gonorrhea, gonococcal arthritis, conjunctivitis, inflammation of the great vestibular gland, and Fitz–Hugh–Curtis syndrome. Gonococcal infection can also lead to PID which is strongly associated with TFI [46]. TFI is also dependent on the number and severity of PID episodes [47]. PID includes inflammation of the endometrium, the ovaries, and the fallopian tubes. Tubal abscesses may also be present.

Research regarding adverse birth outcomes and maternal infections indicates that gonorrhea is also strongly associated with a 60% increased probability of small for gestational age (SGA) infants and a 40% increased probability of low birth weight (LBW) infants. Other possible complications during pregnancy include prematurity and neonatal infection, increased perinatal mortality, premature rupture of membranes, and postpartum endometritis [3] (see Table 1).

### 4.2. Chlamydia trachomatis

*C. trachomatis* infection can be either symptomatic or asymptomatic in women and 70–95% of infected women have no symptoms. The first symptoms usually occur 3–7 weeks after the intercourse. Women’s symptoms usually include an abnormal vaginal discharge and a burning sensation when urinating. Additional symptoms are presented in Figure 1. About 2.5–4.5% of women with chlamydia will develop PID if the chlamydia infection is untreated [48].

PID can be a main cause of infertility, difficulty getting pregnant, or of tubal pregnancy [47]. Therefore, many countries recommend routine screening to identify most infections. Maternal *C. trachomatis* infection can also result in infant conjunctivitis and pneumonia and maternal postpartum endometritis.

### 4.3. Mycoplasma genitalium

Up to 81.9% of patients with *M. genitalium* infection are asymptomatic [3]. If symptoms are present, they usually include urethritis, increased or altered vaginal discharge, bleeding between periods, pain during intercourse, discharge or bleeding after intercourse, and abdominal pain.

*M. genitalium* infection can result in cervicitis, postpartum endometritis, neonatal disease, and reactive arthritis [49].

The role of *M. genitalium* infection in infertility is underestimated. The infection can lead to fallopian tube disorders. It has been proven that *M. genitalium* can be present in up to 16% of infertile women [50]. Adverse pregnancy outcomes include spontaneous abortion, miscarriage, ectopic pregnancy, and recurrent pregnancy loss. Rectal and pharyngeal infections are usually asymptomatic [51].

Neonates of *M. genitalium* positive mothers should be observed for conjunctivitis and respiratory tract infections [51].

## 5. Molecular Diagnostics

The most recommended methods for the laboratory diagnosis of chlamydial infection, *M. genitalium* infection, and gonorrhea are nucleic acid amplification tests (NAATs). Over the last decade, NAATs have replaced traditional diagnostic methods. Molecular methods are more sensitive than culture and less demanding when it comes to the collection, storage, and transportation of clinical specimens. However, in the case of *N. gonorrhoeae,* culture is still recommended because it is an inexpensive method that makes it possible to perform antimicrobial susceptibility tests and epidemiological genotyping of the bacteria [61]. Because the culture of *M. genitalium* and *C. trachomatis* is expensive and time and labor consuming, these methods are not currently used in routine diagnostics. Compared with enzyme immunoassay (EIA) or direct immunofluorescence (DIF), NAATs are not only more sensitive but also more specific. The EIA and DIF tests can be utilized for the diagnosis of *C. trachomatis* only in cases when NAATs are not accessible. In women, urine is not the optimal specimen for NAATs because of the possible occurrence of amplification inhibitors, however, self-collected or physician-collected vaginal swabs are highly acceptable [51,61,62,63].

The most important methods currently used in the molecular diagnosis of *N. gonorrhoeae, C. trachomatis*, and *M. genitalium* infections are real-time PCR, strand displacement amplification (SDA) and transcription-mediated amplification (TMA). There are many commercial tests that can be utilized in the diagnosis of *N. gonorrhoeae* and *C. trachomatis.* FDA-approved tests are presented in Table 2 [62,64].

Although the tests are usually highly specific and sensitive, some false-positive and false-negative results have been described. The reason for false-positive results can be cross-reaction with identical or very similar DNA fragments occurring in other related species. In particular, in the case of *N. gonorrhoeae*, cross-reactions with other *Neisseria* species can occur [65].

False-negative results can be caused by sequence variations in target fragments in the host DNA. The problem can, for example, affect the sensitivity of tests that use cryptic plasmids as a target in the diagnosis of *C. trachomatis* infections. Such tests cannot detect a new variant of *C. trachomatis,* the so-called “Swedish variant”, possessing a 377 BP deletion in the plasmid or plasmid-free *C. trachomatis.* Another variant connected with false-negative results of the Aptima Combo 2 assay was single-nucleotide polymorphism (SNP) in the bacterial gene encoding 23S rRNA (Finnish new variant *C. trachomatis* - nvCT). However, the problem has now been resolved and a new version of the test detects Finnish nvCT [62,63,66]. *N. gonorrhoeae* false-negative results caused by genetic variations of the target site are less common and concerns only some tests produced by laboratories (home-made tests) [64].

Home-made tests for the diagnosis of *C. trachomatis, N. gonorrhoeae*, and *M. genitalium* infections should be enhanced with quality control.

There are many real-time PCR tests, detecting MgPa adhesion protein 66 and G3PDH or 16S rRNA genes of *M. genitalium*. Another method used in the diagnosis of *M. genitalium* infection is transcription-mediated amplification (TMA) detecting the unique fragments of 23SrRNA or 16S rRNA. A problem in the laboratory diagnosis of *M. genitalium* can be the very small number of bacteria in the specimen and this can affect sensitivity [51,64].

The Aptima *Mycoplasma genitalium* Assay from Hologic Inc. is the first FDA-approved test for the diagnosis of *M. genitalium* infection (Table 2). The test detects 16s rRNA of *M. genitalium* and is based on target capture, transcription-mediated amplification (TMA), and hybridization protection assay (HPA) technologies [67]. Because of the high prevalence of the resistance to azithromycin, macrolide resistance tests are recommended before treatment. There are many home-made NAATs that detect mutations in 23S rRNA responsible for resistance, but recently some commercial tests have appeared as well. An interesting solution is the Resistance Plus MG FleXible (RPMG Flex) assay (SpeeDx). This is a multiplex qPCR test detecting the gene encoding the MgPa adhesion protein and simultaneously detects five 23S rRNA mutations (A2058G, T, or C, and A2059G or C) [68].

Another problem is that most NAATs are time consuming. In the case of sexually transmitted infection, the optimal solution would be a fast, specific, and sensitive point of care test. Unfortunately, most of the investigated point of care NAATs had insufficient sensitivity to be approved for use. However, promising results (a sensitivity of 97%) have been achieved with the io^®^ single module system (Atlas Genetics Ltd.), which is a 30-minute point of care test to detect *C. trachomatis* in women [69]. Among the sensitive and specific FDA-approved tests (see Table 2), the Xpert CT/NG is relatively fast (90 min) [70].

## 6. Therapy (with Special Attention to Therapy in Pregnant Women)

### 6.1. Neisseria gonnorhoeae

The new guidelines published by the International Union against Sexually Transmitted Infections (IUSTI) recommend, as a first-line treatment, ceftriaxone IM 1 g in a single dose and azithromycin 2 g in a single oral dose. In cases when NAATs to detect *C. trachomatis* and control tests for gonorrhea are performed, monotherapy with ceftriaxone 1 g is acceptable as well.

Patients for whom injections are contraindicated or who refuse to have injections can receive second-line treatment consisting of cefixime 400 mg plus azithromycin 2 g in single oral doses.

The recommended treatment for patients allergic to cephalosporin includes spectinomycin 2 g IM in a single dose and azithromycin 2 g in a single dose. An alternative treatment for allergic patients is gentamicin 240 mg IM in a single dose and azithromycin 2 g in a single oral dose. After a positive confirmation of *N. gonorrhoeae* sensitivity, ciprofloxacin 500 mg in a single oral dose can be used as well. If adverse gastrointestinal symptoms are anticipated, then azithromycin in all of these treatments can be given in divided doses of 1 g followed by another 1 g after 6–12 hours. In case of confirmed ceftriaxone resistance, the recommended treatment for all patients who received lower doses of the antibiotics or the alternative treatment is ceftriaxone 1 g and azithromycin 2 g. Either spectinomycin and azithromycin or gentamicin and azithromycin, in doses as described above, can be used as well.

According to the guidelines of the Centers for Disease Control and Prevention (CDC), first-line treatment consists of ceftriaxone 250 mg IM in a single dose together with azithromycin 1 g in a single oral dose. Second-line treatment includes cefixime 400 mg in a single oral dose together with azithromycin 1 g in a single oral dose. In case of confirmed *N. gonorrhoeae* resistance to ceftriaxone or a patient’s allergy to the antibiotic, single oral doses of gemifloxacin 320 mg together with azithromycin 2 g in a single oral dose or single doses of gentamicin 240 mg IM together with azithromycin 2 g can be used. Pregnant women should get the same first-line dual therapy protocol.

Spectinomycin 2 g IM in a single dose together with azithromycin 2 g in a single oral dose can be used as an alternative treatment. Aminoglycosides (gentamicin) or quinolones (ciprofloxacin, gemifloxacin) should not be used for the treatment of gonorrhea in pregnant women [45,71].

### 6.2. Chlamydia trachomatis

Uncomplicated *C. trachomatis* infection is usually treated with doxycycline 100 mg twice a day for seven days or azithromycin 1 g orally immediately (according to IUSTI, azithromycin is second-line treatment) [63]. As doxycycline is contraindicated in pregnancy, the first-line treatment for *C. trachomatis* infection in pregnant women is azithromycin 1 g orally.

The second-line treatment includes amoxicillin 500 mg three times a day for seven days orally or erythromycin 500 mg four times a day for seven days orally. The third-line treatment includes josamycin 500 mg three times a day or 1000 mg twice a day for seven days orally.

### 6.3. Mycoplasma genitalium

The first-line treatment of *M. genitalium* infection is azithromycin 500 mg on day one, followed with 250 mg on days 2–5 or josamycin 500 mg for 10 days (orally, three times a day). The second-line treatment is moxifloxacin 400 mg for 7–10 days (orally). The third-line treatment includes doxycycline 100 mg twice a day for 14 days. A cure rate of approximately 90% can be achieved with pristinamycin 1 g four times a day for 10 days. If the infection is complicated with PID then the treatment is moxifloxacin 400 mg once a day for 14 days [51].

Guidelines slightly differ for pregnant women. Azithromycin is the first-choice treatment, but if this treatment cannot be implemented, then some authors consider postponing treatment until after pregnancy. Moxifloxacin is contraindicated in pregnancy and there are no data about using pristinamycin in pregnancy.

## 7. Antibiotic Resistance

### 7.1. Neisseria gonorrhoeae

Escalating antibiotic resistance in *N. gonorrhoeae* represents a significant problem. In 2017, *N. gonorrhoeae* was placed on the WHO global priority list of antibiotic-resistant bacteria which includes 12 pathogens for which new antibiotics are urgently needed. Multiresistant *N. gonorrhoeae* strains, including strains resistant to beta-lactams (ceftriaxone, cefixime, and penicillin) and simultaneously resistant to azithromycin and ciprofloxacin, have appeared all over the world.

The susceptibility/resistance to antibiotics among *N. gonorrhoeae* strains in most European countries is monitored as part of the European Gonococcal Antimicrobial Surveillance Programme (Euro-GASP) [72,73]. In the project, genomes of strains collected from different countries are sequenced with the WGS method, which makes it possible to identify the reasons for their drug resistance and to determine the epidemiologic genotypes using the NG-MAST, NG-STAR, and MLST systems. The sequences of newly described resistance determinants are accessible in the worldwide NG-STAR database [18].

*N. gonorrhoeae* resistance to oxyimino-cephalosporins (ceftriaxone and cefixime) is connected with the combination of mutations in bacterial genes but crucially important are changes in the *penA* gene, which encodes PBP2 transpeptidase. Based on differences in the *penA* gene, a 1745–1752 bp fragment, about 395 alleles of PBP2 have been described. The alleles are divided into 165 main types (1–5, 7, 9–19, 21–22, 27, 34–35, 37–179) and many subtypes. The proteins can differ from each other in one to more than 60 amino acids. The increase in oxyimino-cephalosporin MIC (minimum inhibitory concentration) can also be determined by the overproduction of MtrCDE and amino acid substitutions in the PorB1b protein but neither of the two mechanisms can cause resistance to cephalosporins alone. MtrCDE is a membrane pump protein that removes beta-lactams, macrolides, tetracyclines, rifampicin, and detergents from bacterial cells. Overproduction of MtrCDE is caused by substitutions in the repressor protein regulating the production of MtrCDE (G45D and A39T) and mutations in the promotor region of the *mtrR* gene. The substitution of amino acids in the PorB1b protein at positions 120 and 121 causes changes in the MIC levels of penicillins, cephalosporins, and tetracyclines. The sequencing of 30 bp fragments of the *porB1b* gene revealed the following substitutions in the PorB1b protein: G120C, D, E, K, N, Q, R, S, or T and A121D, G, N, S, or V [74,75].

High-level resistance to penicillin is determined by the production of beta-lactamases encoded by genes located in bacterial plasmids. The most common enzyme is TEM-1 but others, such as TEM-135, TEM-220, TEM-75, TEM-141, and TEM-198, can also appear. Chromosomal resistance to penicillin is caused by a combination of mutations: overproduction of the MtrCDE pump and substitutions of amino acids in the PorB1b (120/121) and PonA (421) proteins. The penicillin MIC is usually lower than in the case of the production of beta-lactamases [76,77].

Resistance to macrolides (azithromycin) is connected with transition in the *rrl* (23S rRNA) gene: A2049G (high-level resistance to azithromycin, MIC > 256 mg/L) or C2611T azithromycin (MIC in the range of 2–16 mg/L in the case of transition in all four alleles of the gene). Overproduction of the membrane pump MtrCDE can cause an azithromycin MIC of about 0.5 mg/L [20].

*N. gonorrhoeae* resistance to quinolones is caused by amino acid substitutions in topoisomerases II and IV GyrA (S91F or T, D95A, G, N, or Y) and ParC (D86N, S87C, I, K, N, R, or Y, S88A or P, E91A, G, K, or Q) [75,76].

Resistance to tetracyclines can be caused by the production of TetM, a protein that actively protects the bacterial ribosome from the action of antibiotics. TetM in *N. gonorrhoeae* is encoded by genes localized in the conjugative plasmids and determines a tetracycline MIC of 16–64 mg/L. Two types of tetracycline plasmids have been identified, which have been named Dutch and American. Chromosomal resistance to tetracyclines (with a tetracycline MIC of 2–4 mg/L) is associated with a combination of the overproduction of the membrane pump MtrCDE and mutations in the *penB, penC*, and *rpsJ* genes [78,79].

### 7.2. Chlamydia trachomatis

*C. trachomatis* is relatively sensitive to antibiotics; however, a few cases of resistance have been described. The reason for *C. trachomatis* resistance to macrolides (azithromycin) can be a mutation in the *rplD* gene that encodes the ribosomal L4 protein (substitution Q66K) connected with 23S rRNA conformational changes of the II, III, and V 23S rRNA domains. Other mutations that can lead to macrolide resistance are transversion A2058C and transition T2611C *(Escherichia coli* numbering) in the peptidyl-transferase region of the 23S rRNA gene and mutations in *rplV* encoding L22 protein (substitutions Q52K, R65C, V77A) [80].

Resistance to tetracyclines in *C. trachomatis* is caused by the production of the Tet(C) efflux pump encoded by genes located in genomic islands inserted into the bacterial chromosome. To date, two genomic islands (6 kb and 13.5 kb) have been described in *C. trachomatis*. The *tetC* gene can usually be found with the *tetR* (regulatory) gene and the insertion sequence IScs605. Resistance to tetracyclines may be transmitted horizontally from *C. suis* to *C. trachomatis* [81].

*C. trachomatis* resistance to quinolones (ciprofloxacin) is associated with a mutation in the *gyrA* gene (substitution S83I in the GyrA protein). Resistance to rifampicin can be caused by a mutation in the *rpoB* gene encoding the beta subunit of DNA-dependent RNA polymerase [82].

### 7.3. Mycoplasma genitalium

*M. genitalium* is inherently resistant to all beta-lactam antibiotics, because of the absence of a cell wall. The bacteria are usually sensitive to tetracyclines in vitro, however, clinical efficacy of doxycycline treatment is only 20–40%. The percentage of strains resistant to macrolides (azithromycin) and quinolones (moxifloxacin) has increased in recent years [83].

*M. genitalium* resistance to macrolides (azithromycin) is connected with point mutations in 2058 and 2059 (*E. coli* numbering) in the V region of 23S rRNA. Resistance to quinolones (moxifloxacin) in this species can be associated with mutations in the topoisomerase IV gene causing substitutions in the ParC protein (especially S83R, S83I, D87N, or D87Y) [84,85].

The data concerning the prevalence of the mutations in different geographic regions are incomplete, but according to accessible information, the overall prevalence of macrolide-resistant strains increased from 10% before 2010 to 51.4% in 2016–2017. The prevalence of moxifloxacin-resistant strains remained the same, at 7.7%, during this period. The percentage of *M. genitalium*-resistant to antibiotics is higher in the South Pacific and the American WHO regions than in the European region [51,86].

## 8. New Antibiotics

### 8.1. Marketing Authorization Already Approved

Delafloxacin, a new fluoroquinolone, indicated for treatment of skin infections and community-acquired pneumonia, demonstrated high in vitro activity against *N. gonorrhoeae*, including ciprofloxacin-resistant strains, and is active against *C. trachomatis*. Clinical studies would be necessary to define the real effectiveness of the antibiotic. Another new antibiotic from this group, sitafloxacin, was effective against *M. genitalium* in vitro and in an investigated group of Japanese patients with cervicitis and urethritis. However, fluoroquinolones are contraindicated in pregnant women and should be used with caution in other patients and not as a first-line treatment, because of reports of serious adverse effects such as hepatotoxicity, tendinopathies, neurological symptoms, arrhythmias, and an increased risk of aortic dissection [83].

Solithromycin is a fluoroketolide which the FDA approved for the treatment of community-acquired pneumonia. The antibiotic inhibits the synthesis of bacterial proteins and has a similar but not identical mechanism of action to the macrolides. Solithromycin binds to bacterial rRNA in three different sites and is usually active against macrolide-resistant bacteria. Solithromycin also shows activity against *N. gonorrhoeae, C. trachomatis*, and *M. genitalium.* However, the phase 3 clinical trial has shown that the monotherapy of gonorrhea with oral solithromycin is less successful (80% negative tests of cure) than the standard treatment with ceftriaxone and azithromycin (84% negative tests of cure) [87].

Lefamulin is a new pleuromutilin antibiotic approved in the United States in 2019 and in Europe in 2020 for the treatment of community-acquired pneumonia. Pleuromutilins inhibit bacterial protein synthesis by binding to the peptidyl transferase center in the 50S bacterial ribosome. Pleuromutulins have been used for many years in veterinary medicine but lefamulin is the first pleuromutilin approved for use in humans. The mechanism of action of lefamulin is different to that of macrolides and the antibiotic is usually active against macrolide-resistant bacteria. The antibiotic showed high in vitro activity against *N. gonorrhoeae, C. trachomatis*, and *M. genitalium*, including antibiotic-resistant strains of *N. gonorrhoeae* and *M. genitalium* [88].

### 8.2. Investigational Drugs

Zoliflodacin (AZD 0914 or ETX 0914, Entasis Therapeutics) is a spiropyrimidinetrione that inhibits bacterial DNA gyrase/topoisomerase using a different mechanism of action than quinolones. Zoliflodacin is active against many bacteria, including *N. gonorrhoeae*, *C. trachomatis*, and *M. genitalium* [89].

Another new bacterial DNA synthesis inhibitor is gepotidacin (GlaxoSmithKline), a triazaacenaphthylene that targets bacterial topoisomerase II. Similar to zoliflodacin, the antibiotic has a mechanism of action distinct from that of fluoroquinolones and is active against different species of bacteria, including sexually transmitted bacteria. Gepotidacin and zoliflodacin have started phase 3 of clinical investigations, including establishing the effectiveness for the treatment of gonorrhea. Phase 2 of clinical trials revealed a good safety profile and effectiveness for the treatment of gonorrhea; however, rare treatment failures were reported in the case of both antibiotics [89,90].

DIS-73285 and *SMT-571* are novel small molecule antimicrobials that exhibit high in vitro activity against *N. gonorrhoeae*, including antibiotic-resistant strains. These antimicrobials have completely new, distinct mechanisms of action. SMT-571 interferes with bacterial cell division and the targets of DIS-73285 are electron transfer proteins (ETFs) A/B/D of *N. gonorrhoeae.* In the case of SMT-571, initial in vivo studies confirmed oral bioavailability. Clinical trials should be able to show whether one or both of these interesting substances will be a new treatment for gonorrhea [91,92].

Apramycin is an aminoglycoside antibiotic produced by *Streptomyces tenebrarius* and is used in veterinary medicine. The antibiotic binds to the bacterial ribosome more selectively than other aminoglycosides, which probably improves safety and especially reduces the possibility of nephrotoxicity and ototoxicity. The antibiotic has demonstrated rapid bactericidal activity against *N. gonorrhoeae* in vitro, including spectinomycin-resistant strains [93].

Aminomethyl spectinomycins (*N*-benzyl-substituted 3′-(*R*)-3′-aminomethyl-3′-hydroxy spectinomycin) are new spectinomycin analogs. The antibiotics are in vitro active against many respiratory tract pathogens and against *N. gonorrhoeae*, *C. trachomatis*, and *M. genitalium* [94].

## 9. Conclusions

The number of STIs is rising worldwide. Screening programs (using sensitive and specific diagnostic methods), fast and effective treatment of detected cases, and proper prophylaxis should be implemented to avoid possible complications due to STIs such as fertility-related issues and adverse pregnancy outcomes. It may be important to track epidemiological outbreaks and the spread of individual strains using molecular methods.

The rapid spread of antibiotic-resistant *N. gonorrhoeae* and *M. genitalium* necessitates the monitoring of the problem and the development of new drugs and treatment regimens.

## Figures and Tables

**Figure 1 ijms-22-02170-f001:**
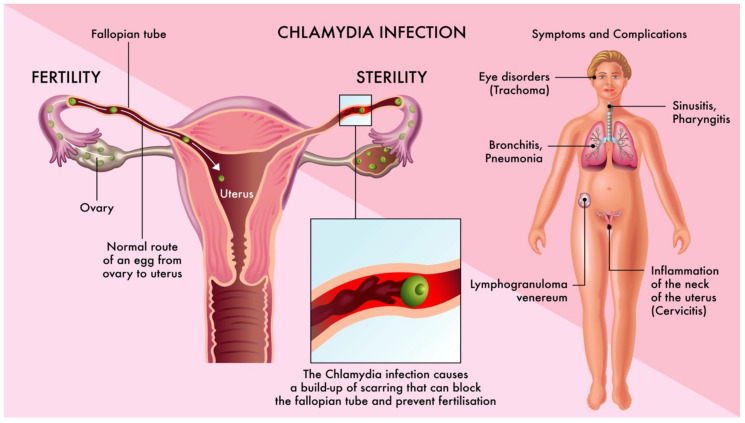
*Chlamydia trachomatis* infection. Symptoms and possible impact on fertility.

**Table 1 ijms-22-02170-t001:** Adverse pregnancy outcomes regarding *N. gonorrhoeae*, *C. trachomatis*, and *M. genitalium* infections.

***Neisseria gonorrhoeae***	***Chlamydia trachomatis***	***Mycoplasma genitalium***
Ectopic pregnancy ⇐⇐ pelvic inflammatory disease (PID) ⇒⇒ infertility [3,44,52]		
Miscarriage [53]		
Increased perinatal mortality	Increased perinatal mortality [54]	Recurrent pregnancy loss [53]
Neonatal infection (conjunctivitis)	Neonatal infection (conjunctivitis, pneumonia)	Neonatal infection [55]
Postpartum endometritis [56]	Postpartum endometritis –ve [54]/+ve [57]	Postpartum endometritis
Low birth weight (LBW) [43]		
Preterm birth [44,52]		
Small for gestational age (SGA) [43]	Not small for gestational age (SGA) [58]	Not small for gestational age (SGA) [59]
Premature rupture of membranes [52,60]		

**Table 2 ijms-22-02170-t002:** FDA-approved tests utilized in the diagnosis of NG- *N. gonorrhoeae,* CT *-C. trachomatis* and MG *-M. genitalium*. SDA: strand displacement amplification, TMA: transcription-mediated amplification, ORF- open reading frame.

NAAT	Producer	Detected Pathogen	Methods Used	Target
**Aptima Combo 2 assay**	Hologic Inc.	CT/NG	TMA	NG, specific regions in 16S rRNA CT, specific regions in 23Sr RNA
**Aptima CT assay**	Hologic Inc.	CT	TMA	Specific regions in 16S rRNA
**Aptima NG assay**	Hologic Inc.	NG	TMA	Specific regions in 16S rRNA
**Abbott RealTime CT/NG**	Abbott	CT/NG	real-time PCR	CT, two specific regions in cryptic plasmid NG, specific sequence in OPA gene
**cobas 4800 CT/NG Test**	Roche Diagnostics	CT/NG	PCR	CT, two targets, one in cryptic plasmid and one in chromosome NG, direct repeat (DR) 9 specific regions
**ProbeTec ET CT/GC Amplified DNA assay**	Becton Dickinson	CT/NG	SDA	CT, specific region in cryptic plasmid (ORF) NG, pilin gene inverting protein homolog
**ProbeTec CT Q^X^ Amplified DNA assay**	Becton Dickinson	CT	SDA	Specific region in cryptic plasmid (ORF)
**ProbeTec NG Q^X^ Amplified DNA assay**	Becton Dickinson	NG	SDA	Specific region in pilin gene
**Xpert CT/NG**	Cepheid	CT/NG	real-time PCR	CT, specific chromosomal DNA sequence NG, two specific chromosomal DNA sequences (both should be detected for a positive result)
**Aptima *Mycoplasma genitalium* Assay**	Hologic Inc.	MG	TMA	Specific regions in 16s rRNA

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
