# Peer review of "The Impact of Selected Bacterial Sexually Transmitted Diseases on Pregnancy and Female Fertility"

_ijms, 2021, doi:10.3390/ijms22042170_

Round 1

Reviewer 1 Report

The review article provides some significant contributions to the field of the infection-associated complications afflicting the female reproductive tract; namely by providing an overview and description of molecular diagnostic techniques, resistance and therapy innovations as it pertains to Chlamydia trachomatis, Neisseria gonorrhoeae and Mycoplasma genitalium. As these infections are still prevalent, recurrent (any significant/ life-long immunity does not establish), potentially cause morbidities and their multi-resistant strains are spreading globally (N. gonorrhoeae), such review can be of significance to the clinical and scientific community and the public in general.
However, I would like to outline what I find to be some major deficiencies.

The main points that characterise the manuscript all throughout:
1. Lack of clarity on the scope of the review. The challenges I see would perhaps become a bit more explicit by trying to answer these questions:

a) Is the focus on fertility issues and tubal factor infertility?

b) Or is the goal of the review to provide a needed update on molecular diagnostics and epidemiology of these infections, as well as therapies in the pipeline? (in other words: Is the review the first in this field to provide an overview of these topics? Or an update? What is the novelty it brings?)

c) If the authors’ vision was to provide the latter to the field (described in b), how do the other chapters assist with this goal?

2. Insufficient coordination between the authors. Every section feels uncoupled from the other ones, in a way that it reads as if each author wrote 1-2 chapters and everything got complied into one document at the end without editing, creating an incoherent manuscript without a flow, or start and finish.

3. Questionable relevance of the pathophysiology and symptomatology sections. Are they meant as an adjunct to the previously mentioned sections? They do not seem to bring novelty to the manuscript, all cited data has been around for some time. E.g. the pathology of women section does not feature practically any studies post 2010. There is a latest study in the pregnancy section, but how is that relevant when assessed together with the diagnostics? Do new technologies enhance screening efforts and reduce the infection rates and/or complications? And if so, what are the latest stats? The manuscript, as it is, opens more new questions than it answers.

4. Consistency

Abbreviations: The names of different bacteria are inconsistently abbreviated. In almost every chapter abbreviations are introduced anew, only to be ignored few sentences down the line by writing the full name. It again appears as if there was lack of communication between different authors when they contributed with their own sections, as well as insufficient proof-reading before submission.

Epidemiology: The stats on the incidence and/or prevalence of these infections are quite inconsistent. The numbers are given from 2017 for one infection, but from 2006 for another. There should be more recent published data on these infections than 2006.

Language: The level of English is solid, it is easy to understand the authors’ points. However, it might still benefit from one round of proof-reading. There are instances where the language is e.g. insufficiently clear, here are only a few examples:

l.412: Currently registered (new antibiotics) – It can sound ambiguous to a reader: currently in the process of getting registered (marketing authorization submitted), or already registered? I am fairly sure you meant the latter, but please make it explicit, e.g. “Marketing authorization already approved”.

l.63 “analyse” where “analysis” is meant.

5. Organisation/structure

Abstract: It does not reflect fully the focus of this review. The review covers several topics within STI-associated fertility issues,

Conclusions: It is quite brief and does not reflect well the content of the review. I would advise that you first decide as a group how you would like to make the aim and scope of the review better defined and focused (see suggestions above), and then rewrite this section entirely to reflect this new, crystalized focus.

If you choose to go for the molecular diagnostics as the primary focus, then either bring it back to its significance for the fertility issues (Will they help screening efforts/prevention? Will they help in timely treatment, reduce the impact of these infections on women’s reproductive health/PID/TFI?).
It would help you prepare a well-rounded manuscript.

Other remarks:

1. You refer to these as STDs throughout the entire manuscript, but nowadays the majority of the field refers to them as sexually transmitted infections (STIs). These infections do not always lead to a disease and often enough they are asymptomatic. Therefore the name STD can be misleading, so I would advise you reconsider this.

2. I did not detect excessive self-referencing, aside from three publications by Mlynarczyk-Bonikowska. Since the author does publish on the topic, please explain whether those are the only sources of that data and must therefore be included.

In conclusion: It appears to me personally that the greatest strength of your review would be the contribution to the topic of molecular diagnostics/epidemiology, resistance and novel therapies, because these chapters are the most robust in their writing and bring the greatest value and novelty in the context of a literature review. The pathology and symptomatology of these infections are relevant issues, however they feel out of place here, are already addressed in other reviews and are not updated here. Hence the manuscript needs to be rewritten based on the newly agreed topic, aims and breadth.
There are remarks in regards to the cohesion of the manuscript chapter-to-chapter, as well as consistency and its structuring and organization. In the light of these, I advise a major revision before a re-consideration can be made.

Author Response

Dear Reviewers

Dear Editors,

We attach the corrected manuscript with the following changes:

REVIEWER 1:

The review article provides some significant contributions to the field of the infection-associated complications afflicting the female reproductive tract; namely by providing an overview and description of molecular diagnostic techniques, resistance and therapy innovations as it pertains to Chlamydia trachomatis, Neisseria gonorrhoeae and Mycoplasma genitalium. As these infections are still prevalent, recurrent (any significant/ life-long immunity does not establish), potentially cause morbidities and their multi-resistant strains are spreading globally (N. gonorrhoeae), such review can be of significance to the clinical and scientific community and the public in general.
However, I would like to outline what I find to be some major deficiencies.

The main points that characterise the manuscript all throughout:
1. Lack of clarity on the scope of the review. The challenges I see would perhaps become a bit more explicit by trying to answer these questions:

  1. a) Is the focus on fertility issues and tubal factor infertility?
  2. b) Or is the goal of the review to provide a needed update on molecular diagnostics and epidemiology of these infections, as well as therapies in the pipeline? (in other words: Is the review the first in this field to provide an overview of these topics? Or an update? What is the novelty it brings?)
  3. c) If the authors’ vision was to provide the latter to the field (described in b), how do the other chapters assist with this goal?

Response: You are right. Our vision was to provide rather an update on the molecular diagnostics and epidemiology of these infections as well as therapies in the pipeline. The goal of the other chapters was to present the importance of the subject and they form an important part of the paper. In accordance with your comments, we have changed the abstract and the introduction part.

2. Insufficient coordination between the authors. Every section feels uncoupled from the other ones, in a way that it reads as if each author wrote 1-2 chapters and everything got complied into one document at the end without editing, creating an incoherent manuscript without a flow, or start and finish.

Response: We reedited the manuscript

  1. Questionable relevance of the pathophysiology and symptomatology sections. Are they meant as an adjunct to the previously mentioned sections? They do not seem to bring novelty to the manuscript, all cited data has been around for some time. E.g. the pathology of women section does not feature practically any studies post 2010. There is a latest study in the pregnancy section, but how is that relevant when assessed together with the diagnostics? Do new technologies enhance screening efforts and reduce the infection rates and/or complications? And if so, what are the latest stats? The manuscript, as it is, opens more new questions than it answers.

Response: We modified the chapter and added newer papers to the literature

4. Consistency

Abbreviations: The names of different bacteria are inconsistently abbreviated. In almost every chapter abbreviations are introduced anew, only to be ignored few sentences down the line by writing the full name. It again appears as if there was lack of communication between different authors when they contributed with their own sections, as well as insufficient proof-reading before submission.

Response: We corrected it in all of the text.

Epidemiology: The stats on the incidence and/or prevalence of these infections are quite inconsistent. The numbers are given from 2017 for one infection, but from 2006 for another. There should be more recent published data on these infections than 2006.

Response: We added 2018 data. Differences in the years from which data were given are due to different availability of the data.

Language: The level of English is solid, it is easy to understand the authors’ points. However, it might still benefit from one round of proof-reading. There are instances where the language is e.g. insufficiently clear, here are only a few examples:

l.412: Currently registered (new antibiotics) – It can sound ambiguous to a reader: currently in the process of getting registered (marketing authorization submitted), or already registered? I am fairly sure you meant the latter, but please make it explicit, e.g. “Marketing authorization already approved”.

Response: Yes, it means that marketing authorization was already approved and the sentence has been corrected according to your suggestion.

l.63 “analyse” where “analysis” is meant.

Response: Our mistake, this has been corrected. The article currently has undergone a second proofreading.

5. Organisation/structure

Abstract: It does not reflect fully the focus of this review. The review covers several topics within STI-associated fertility issues,

Response: The abstract was corrected. The focus of the review has been added.

Conclusions: It is quite brief and does not reflect well the content of the review. I would advise that you first decide as a group how you would like to make the aim and scope of the review better defined and focused (see suggestions above), and then rewrite this section entirely to reflect this new, crystalized focus.

Response: The abstract was corrected. The focus of the review has been added.

If you choose to go for the molecular diagnostics as the primary focus, then either bring it back to its significance for the fertility issues (Will they help screening efforts/prevention? Will they help in timely treatment, reduce the impact of these infections on women’s reproductive health/PID/TFI?).
It would help you prepare a well-rounded manuscript. Te

Response: In the chapter “introduction”, it was underlined that STIs have a great impact on human reproduction and pregnancy and that early diagnosis and treatment are crucial.

Other remarks:

1. You refer to these as STDs throughout the entire manuscript, but nowadays the majority of the field refers to them as sexually transmitted infections (STIs). These infections do not always lead to a disease and often enough they are asymptomatic. Therefore, the name STD can be misleading, so I would advise you reconsider this.

Response: We changed the abbreviation to STI.

2. I did not detect excessive self-referencing, aside from three publications by Mlynarczyk-Bonikowska. Since the author does publish on the topic, please explain whether those are the only sources of that data and must therefore be included.

Response:  We consider these publications valuable, but considering your opinion, we replaced two of them with papers by other authors.

In conclusion: It appears to me personally that the greatest strength of your review would be the contribution to the topic of molecular diagnostics/epidemiology, resistance and novel therapies, because these chapters are the most robust in their writing and bring the greatest value and novelty in the context of a literature review. The pathology and symptomatology of these infections are relevant issues; however they feel out of place here, are already addressed in other reviews and are not updated here. Hence the manuscript needs to be rewritten based on the newly agreed topic, aims and breadth.
There are remarks in regard to the cohesion of the manuscript chapter-to-chapter, as well as consistency and its structuring and organization. In the light of these, I advise a major revision before a re-consideration can be made.

Response: Thank you for the review. We trust that we have managed to correct our manuscript and that it is now acceptable.

Reviewer 2 Report

The manuscript is interesting in the field of epidemiology, pathogenesis and diagnostics of STD bacteria. The manuscript should be completed by data on molecular aspects of epidemiology and pathogenesis, e.g. data on pathogen-host cells interactions, molecular biology of STD bacteria. 

Author Response

Dear Reviewers

Dear Editors,

We attach the corrected manuscript with the following changes:

REVIEWER 2:

The manuscript is interesting in the field of epidemiology, pathogenesis and diagnostics of STD bacteria. The manuscript should be completed by data on molecular aspects of epidemiology and pathogenesis, e.g. data on pathogen-host cells interactions, molecular biology of STD bacteria.

Response: Thank you for review. Your suggestions were considered. We have added some text about molecular aspects of pathogenesis and literature. Molecular epidemiology was a subject of the separate chapter entitled “Genotyping and molecular epidemiology”, next to the chapter “Epidemiology”.

Round 2

Reviewer 1 Report

The manuscript has been substantially improved. There are more minor spell checks left to complete.
Moreover, the terminology for the infections has been standardised (e.g. C.trachomatis). However, there are still occasional usages of "chlamydia" and "CT" in the text. These minor mistakes still ought to be corrected. I expect this can be arranged via the editorial office.

Author Response

Thank you for the review. We corrected the usage of chlamydia and ct in the text. 

Reviewer 2 Report

In my opinion, the manuscript was not fundamentally improved. Although, the sentence “This paper presents the latest data on molecular diagnostics, current and new treatments, and drug resistance.” was added, the subject was not sufficient explained.

Author Response

Thank you for the review. We modified the manuscript, added the recent data.

Round 3

Reviewer 2 Report

The Authors added only few sentences.

Author Response

Thank you for the review. We took into consideration all the reviewer's remarks as well as editor's notes. We updated the molecular epidemiology section, added one figure. 

Moreover, we  found and corrected 1 typo and replaced all CTs in the text with C. trachomatis and NG with N. gonorrhoeae.